# Protein Characteristics and Bioactivity of Fish Protein Hydrolysates from Tra Catfish (*Pangasius hypophthalmus*) Side Stream Isolates

**DOI:** 10.3390/foods11244102

**Published:** 2022-12-19

**Authors:** Hang Thi Nguyen, Huynh Nguyen Duy Bao, Huong Thi Thu Dang, Tumi Tómasson, Sigurjón Arason, María Gudjónsdóttir

**Affiliations:** 1Faculty of Food Science and Nutrition, University of Iceland, Aragata 14, 102 Reykjavik, Iceland; 2Faculty of Food Technology, Nha Trang University, 02 Nguyen Dinh Chieu, Nha Trang 650000, Vietnam; 3UNESCO GRÓ—Fisheries Training Programme, Fornubudum 5, 220 Hafnarfjordur, Iceland; 4Matis, Icelandic Food and Biotech R&D, Vinlandsleid 12, 113 Reykjavik, Iceland

**Keywords:** enzymatic hydrolysis, dark muscle, side streams, antioxidant, fish protein isolates, amino acids

## Abstract

Enzymatic hydrolysis is a novel method to recover highly potent bioactive fish protein hydrolysates (FPHs) from fish processing side-streams. The common way of producing FPHs directly from fish side-streams may be inappropriate due to the excess of lipids and pro-oxidants, especially in lipid-rich streams, as obtained from Tra catfish. This study aimed to optimise the hydrolysis conditions for a commercial enzyme (Alcalase® 2.4 L) (enzyme concentrate, temperature, and time) in FPH production from the fish protein isolate obtained from Tra catfish dark muscle (DM-FPI) using the pH-shift method. The degree of hydrolysis (DH), protein recovery (PR), and antioxidant properties, including DPPH radical scavenging activity (DPPH-RSA) and total reducing power capacity (TRPC), were measured to evaluate the effects of the hydrolysis conditions on the FPHs. Optimal hydrolysis was obtained at an enzyme/substrate protein ratio of 3% (*v*/*w*) and a hydrolysis temperature of 50 °C for 3 h. The FPHs obtained from different substrates, including DM-FPI, abdominal cut-off (ACO) FPI, and head and backbone blend (HBB) FPI, had similar DHs under these optimum conditions, ranging from 22.5% to 24.0%. However, the FPH obtained from abdominal cut-off isolate (ACO-FPH) showed the highest PR of 81.5 ± 4.3% and the highest antioxidant properties, with a DPPH-RSA of 86.1 ± 1.6% and a TRPC of 6.4 ± 0.4 equivalent mg vitamin C/g protein. The resulting FPHs present a natural source of antioxidants with great potential for food applications, especially the ACO-FPH. In addition, all FPHs had excellent amino acid profiles, indicating strong potential for their use as supplements. Tra catfish protein-rich side-streams can thus be processed into high-value bioactive FPHs using Alcalase for human consumption.

## 1. Introduction

Fish has long been recognised as a valuable and nutritious food. Fish is low in saturated fatty acids and cholesterol compared to other animal-origin foods, such as meat, poultry, and eggs. Fish and fish products have been recommended as an important part of a healthy diet, especially if they replace other protein-rich foods with high contents of saturated lipids and cholesterol. Fish and seafood provide substantial essential nutrients, especially proteins, polyunsaturated fatty acids, and minerals [1,2,3]. Several studies have shown that the consumption of fish-derived proteins, such as peptides, fish protein hydrolysates (FPHs), and fish protein isolates (FPIs), also brings more health benefits than consuming intact fish proteins due to their high absorption and digestion properties [2,4].

In recent decades, increasing urbanization among the world population, as well as increased consumer awareness of the health benefits of fish consumption, have led to an increased demand for fish and seafood products [5,6]. In 2019, fish provided about 17% of the total animal proteins and 7% of the total proteins consumed worldwide [7]. Food security issues, especially the supply of sufficient high-quality, nutrient-rich proteins, are a challenge due to the increased growth in the global population and overexploited marine resources [8,9]. Moreover, processing of raw materials from the fishing and fish-farming industries generates substantial underutilized side-streams, many of which have high biological and nutritional value [10,11]. Side streams from fish processing are considered to be all the processing streams that do not contribute to the main production, such as fillets or whole and gutted fish. In developing countries, such as Vietnam, these materials either go to waste or are converted into animal feed, fishmeal, and fertilizer, leading to the underutilization of available resources [12,13]. The Tra catfish is a freshwater species commonly farmed in Vietnam, Thailand, Indonesia, India, and Bangladesh. It is currently the most important farmed freshwater species in Vietnam [7,14]. Tra catfish have a relatively low market price, but their fillets are of good quality, with high protein (18.9%) and low lipid content (2.6%), and the taste is comparable to other whitefish species, such as cod or haddock [15,16]. Frozen white fillets are the most common product of Tra catfish production in Vietnam, resulting in the creation of significant amounts of side streams from the filleting processing [12]. In the case of Tra catfish production, common side-streams originate from the removal of the head and backbones, viscera, dark muscle, and abdominal cut-offs. Fish side-streams often contain high-quality proteins that may be utilized to develop protein-containing foods beneficial for human consumption whilst enhancing economic and environmental sustainability. It is necessary to investigate the potential of converting these fish side-streams into value-added products rather than only producing low-value products, such as fishmeal, fish oil, silage, and fertilizers, as is traditionally the case.

Enzymatic modification of proteins has been applied widely in the food industry for a long time [17,18]. Enzymes help improve the nutritional value of various foods by splitting complex proteins, fats, and carbohydrates into smaller and simpler compounds, increasing their digestibility, absorption, and metabolism [17]. Fish protein hydrolysates (FPHs) are fish-derived proteins containing mixtures of peptides and amino acids of various molecular weights depending on the extent of the enzymes’ hydrolysis. FPHs were among the most important novelty fishery products in the last decade [19]. FPHs enhance the functional aspects of proteins and can be used as additives and/or as ingredients in food applications. Their surface properties, such as hydrophobic and hydrophilic surface groups, can stabilise oil-in-water emulsion and increase foaming activity and water-holding capacity [3,11,20]. FPHs with antioxidant activities are used widely in various industries, such as in health food, aquaculture, pharmaceuticals, cosmetics, and food processing/preservation industries [21,22,23,24,25,26]. In intensive farming systems, fish are kept at high densities, which may increase stress and cause increased susceptibility to diseases, resulting in economic production losses [27]. FPHs and bioactive peptides have, on the other hand, been shown to have positive effects on fish health, immunity, and growth during farming [28,29,30].

FPHs can be produced using different methods, including autolysis, bacterial fermentation, or chemical and enzymatic hydrolysis [3,31,32]. Enzymatic hydrolysis is considered the most efficient method to recover protein hydrolysates from fish side-streams [18,33]. By using different proteolytic enzymes, it may be possible to produce a broad spectrum of food ingredients for a wide range of applications. Enzymatic hydrolysis generates precise hydrolysates, retaining the nutritive value of the source protein with high protein recovery in a short time, and it is beneficial for targeting specific derived products [18]. Several authors have studied the enzymatic proteolysis and solubilization of proteins from different types of fish side-streams in recent decades [18,23,33,34]. In particular, several FPHs with antioxidant activities have been prepared from side streams of fish species, such as Pacific hake [32], round scad [34], cod [35], hoki, pollock, sandfish, tilapia [23], and rainbow trout [36]. Proteases from various sources, such as from animals, plants, bacteria, and fungi, have been used during FPH production. Alcalase^®^ is a commercially available alkaline bacterial protease produced from *Bacillus licheniformis*. This enzyme has been highly recommended as one of the most effective enzymes to produce FPHs with high yield and high antioxidant activities [14,37,38,39].

Traditionally, FPHs are produced directly from fish processing side-streams, such as the heads, frames, dark muscles, cut-offs, and viscera, either as combined streams or separated streams. The raw materials are minced and homogenized with water and then enzymes are added, after which the hydrolysis process is initiated under the given working conditions [18]. However, high lipid content and the presence of pro-oxidants, such as haemoglobin and myoglobin from the blood, in the side streams may cause rancid, fishy odours in the final product, which limits the further use of the FPHs [40,41,42]. This can be especially challenging during utilization of Tra catfish side-streams, which often have high lipid and ash content ranging from 15.3 to 29.8% and 2.4 to 5.7%, respectively [14]. Therefore, pre-treatment of protein substrates, such as defatting, washing, and/or centrifuging, are necessary to remove the excess lipid and pro-oxidants before using them for FPH production [18,43,44]. Alternatively the FPH can be stabilized by adding antioxidants [45]. Producing FPH from fish protein isolates (FPIs) may thus both reduce unwanted components and improve quality.

The pH-shift method is commonly used to recover proteins from fish side-streams. The proteins are solubilised at either low or high pH (≤3.5 or ≥10.5), then fat and other impurities are removed using high-speed centrifugation, after which the proteins are collected following precipitation at their isoelectric point (pH = 5.5) [41,46]. During FPI production using the pH-shift method, most lipids, ash, and contaminates are removed, resulting in high-quality raw material for further FPH production [12,42]. Kakko et al. [47] found that proteins isolated with the pH-shift method had a higher nutrient value than the enzymatically extracted hydrolysates from the same raw material. Nisov et al. [48] also found that the pH-shift method resulted in higher protein recovery yield than the enzymatic method when treating Baltic herring (*Clupea harengus membras*) and roach (*Rutilus rutilus*).

The current study aimed to investigate the feasibility of obtaining functional fish protein hydrolysates (FPHs) by enzymatically hydrolysing FPIs recovered from protein-rich side-streams from Tra catfish (*Pangasius hypophthalmus*) fillet processing, including the dark muscle (DM-FPI), abdominal cut-offs (ACO-FPI), and a head and backbone blend (HBB-FPI). The hydrolysis conditions for the FPH production from FPI-DM was optimised using a one-factor-in-a-time method [14]. The effects of varying enzyme-substrate ratios (trial 1), hydrolysis temperatures (trial 2), and hydrolysis times (trial 3) on the obtained FPHs were evaluated based on their degrees of hydrolysis, protein recoveries, and antioxidant activities. Furthermore, the amino acid composition and the properties of the FPHs were analysed and compared.

## 2. Materials and Methods

### 2.1. Raw Materials and Sampling

#### 2.1.1. Preparation of Fish Protein Isolates (FPIs) 

Dark muscle (DM), a head and backbone blend (HBB), and abdominal cut-offs (ACO) were collected from industrial Tra catfish fillet processing (Nam Viet, Can Tho, Vietnam), as described in detail by Nguyen et al. [12]. The corresponding FPIs (i.e., DM-FPI, HBB-FPI, and ACO-FPI) obtained from these side streams were prepared according to optimised conditions as described by Nguyen, et al. [12]. The raw material was minced and then homogenised in distilled water at pH 12 and a water-to-raw-material ratio of 8 (volume/weight) for 1 min. The mixture was left in a fridge at 0−4 °C for 150 min for protein solubilization. The homogenate was then centrifuged at 3000 rpm for 20 min at 4 °C (MF 600, Biobiz, Incheon, Korea), and the middle fraction, containing soluble proteins, was recovered. This fraction was adjusted to a pH of 5.5 to precipitate the proteins (Portavo 904×, Knick, Berlin, Germany). The aggregated precipitates were then filtered using a nylon monofilament bag (mesh size: 25 micron; Dong Son Ltd., Ho Chi Minh City, Vietnam), and the obtained proteins were washed with distilled deionized water to a neutral pH and centrifuged at 3000 rpm for 20 min to dewater them, remove the NaCl, and form the fish protein isolates (FPIs). Each obtained FPI was mixed evenly, divided, and weighed into 50 g samples, which were then packed in sealed polyethylene bags and stored at −25 ± 1 °C until use.

#### 2.1.2. Preparation of Fish Protein Hydrolysate (FPH) from DM-FPI

The preparation of FPH through the hydrolysis of the DM-FPI using the Alcalase^®^ enzyme was carried out with different enzyme ratios, temperatures, and durations to obtain the optimal conditions for FPH production. A flow chart of the optimization trials for FPH production from the DM-FPI is shown in Figure 1.

First, the enzyme–substrate protein ratio (volume/weight, %) was optimized during trial 1. About 50 g of DM-FPI sample was added to distilled water to obtain a protein substrate concentration of 7.5% [37]. The pH of the mixture was adjusted to 7.5 using 0.1 N NaOH and 0.1 N HCl ((Portavo 904×, Knick, Berlin, Germany). Different ratios of enzyme, including 0.5, 1.0, 1.5, 2.0, 2.5, 3.0, 3.5, 4.0, and 5.0%, were then mixed with the 7.5% protein DM-FPI mixture. The samples were then incubated at 50 °C for 3 h in a shaking water bath for the hydrolytic reaction (VS-1205SW1, Vision Bionex, Bucheon-si, Gyeonggi-do, South Korea). After hydrolysis, the mixture was heated and incubated at 85 °C for 15 min to terminate the enzyme activity. The heated mixture was then centrifuged and separated into two fractions: a protein solution (fish protein hydrolysate (FPH)) and a sediment (residual FPI). The degree of hydrolysis (DH), protein recovery (PR), antioxidant activities (including 2,2-diphenyl-1-picrylhydrazyl radical scavenging activity (DPPH-RSA)), and total reducing power capacity (TRPC) of the FPHs were determined to evaluate the effects of enzyme ratios on the yield and bioactive properties of the FPH, as described in Section 2.2.

An enzyme ratio of 3% was found to be optimal for protein hydrolysis and, thus, was used in further trials. The FPHs were then prepared at different hydrolysis temperatures, including 45 °C, 50 °C, 55 °C, 60 °C, 65 °C, and 70 °C, in trial 2. The DH, PR, DPPH-RSA, and TRPC were evaluated as above.

Trial 2 indicated that the hydrolysis temperature of 50 °C was optimal for FPH production and it was selected for the subsequent trials. The FPH was then produced with different hydrolysis times, including 1.5, 3, 4.5, 6, 7.5, and 9 h, in trial 3. The DH, PR, DPPH-RSA, and TRPC were determined with the same procedures as mentioned above. Each experiment was carried out in triplicate.

#### 2.1.3. FPH Preparation from Different FPIs at the Optimal Conditions

The HBB-FPHs and ACO-FPHs were produced from the corresponding HBB-FPIs and ACO-FPIs with the optimal procedure obtained in this study. The DH, PR, DPPH-RSA, TRPC, and amino acid profile of each FPH were measured and compared.

#### 2.1.4. Chemicals

All chemicals used in the study were of analytical grade and purchased from Sigma-Aldrich Company (Missouri, TX, USA) and Merck (Darmstadt, Germany).

Alcalase^®^ 2.4 L was purchased from Sigma-Aldrich (Missouri, TX, USA). This protease product was obtained from *Bacillus licheniformis*, Subtilisin A. The Alcalase^®^ was stored at 4 ± 1 °C until use.

### 2.2. Analyses

#### 2.2.1. Proximate Composition of the Tra Catfish Side-Streams and their Corresponding FPIs

Water content was determined according to ISO 6496:1999 [49]. About 5.0 g of sample was weighed in a small porcelain bowl. The bowl was dried in an oven for 4 h at 103 ± 1 °C and then allowed to cool to ambient temperature for about 30 min in a desiccator before being weighed.

The crude protein content (total nitrogen content) was measured using the Kjeldahl method according to ISO 5983-2:2009 [50]. Approximately 2 g of minced sample was digested in 17.5 mL concentrated H_2_SO_4_ with two Kjeldahl tablets (each tablet included 3.5 g K_2_SO_4_ and 0.4 g CuSO_4_) as a catalyst at 420 °C for 2.5 h. The digested mixture was made alkaline with NaOH and the nitrogen distilled off as NH_3_. The NH_3_ was “trapped” in a 1% boric acid solution. The amount of ammonia nitrogen in the solution was quantified using a standardized H_2_SO_4_ solution by titration. A nitrogen conversion factor of 6.25 was used to calculate crude protein content.

Lipids were analysed according to the Bligh and Dyer [51] method. Approximately 25 g of the sample was homogenized for 4 min with 50 mL of chloroform, 50 mL of methanol, and 25 mL of 0.88% KCl. The homogenized sample was centrifuged at 2500 rpm for 20 min at 4 °C. The chloroform phase containing lipids (the liquid bottom part) was collected and filtrated through a glass microfiber filter under vacuum suction. Exactly 2 mL of the chloroform fraction was transferred into a glass tube and placed in a vacuum dryer at 55 °C to remove the chloroform solvents. The remaining sample was weighed to calculate the total lipid content.

The ash content was determined according to the method described by the Association of Official Analytical Chemists (AOAC, 2000) [52]. About 5 g of sample was placed into a crucible. Each sample was heated overnight at 550 ± 3 °C and then cooled down in a desiccator before being weighed. Ashes were quantified gravimetrically. The water, crude protein, lipid, and ash contents were expressed as percentages of wet weight (ww).

#### 2.2.2. Amino Acid Analysis

The total amino acid profiles of the FPH samples were determined using the liquid chromatography–mass spectrometry (LC-MS) method according to ISO 13903:2005 [53]. Approximately 1 g of sample was hydrolysed for 24 h in 25 mL of 6 N HCl at 110 °C in a sealed vessel. Amino acids were extracted from the sample using an EZ:faast LC-MS kit system (Phenomenex, Torrance, CA, USA). Ion-exchange chromatography was used to separate the individual amino acids through the EZ:faast AAA-MS column (250 mm × 2.0 mm, 4 µm) (Phenomenex, Torrance, CA, USA). The amino acids were then detected using a ninhydrin reaction at λ 570 nm (λ 440 nm for proline) in a Shimadzu LC-MS 8030 system (Kyoto, Japan). Amino acid content was expressed as g/100 g crude protein.

#### 2.2.3. Degree of Hydrolysis (DH)

The degree of hydrolysis (DH) is defined as the ratio between the number of broken peptide bonds (*h*) and the total number of peptide bonds per mass unit (h*_tot_*) [54], as described by Equation (1):(1)$$\mathrm{DH}\;(\%)=\frac{h}{h_{tot}}\times 100$$
h was determined by measuring the amount of free α-amino group formed in the hydrolysed protein products. The method is based on the formation of a yellow complex between the amino groups in the amino acids with a dinitrofluorobenzene (DNFB) reagent. The absorbance of the solution was read at 410 nm in a DR6000 UV–Vis spectrophotometer (HACH, Düsseldorf, Germany). h*_tot_* is the number of peptide bonds, with 8.6 mol peptide equivalent/kg for fish protein [55]. The equation can then be rewritten as:(2)$$\mathrm{DH}\;(\%)=\frac{A\;\times \;0.001\;\times \;\mathrm{dilution}\;\mathrm{factor}}{P\;\times \;8.6\;\times \;0.001}\times 100$$

A × 0.001 indicates the amount of amino groups (mol/mL) formed based on a standard curve made with glycine with concentrations ranging from 0.0002 to 0.001 mM/mL, and P is the total protein content (g) in 1 g of the hydrolysate solution sample.

#### 2.2.4. Protein Recovery

The protein recovery (PR) from the FPHs was calculated using the following Equation (3):(3)$$\mathrm{PR}\;(\%)=\frac{\mathrm{Total}\;\mathrm{protein}\;\mathrm{in}\;\mathrm{the}\;\mathrm{FPH}}{\mathrm{Total}\;\mathrm{protein}\;\mathrm{in}\;\mathrm{the}\;\mathrm{initial}\;\mathrm{substrate}\;(\mathrm{FPI})}\times 100$$

The protein content of the solution fraction of the FPHs was determined using the Bradford method [56]. For this procedure, 50 μL of the sample was mixed with 2.5 mL of the Bradford reactive solution and then incubated for 25 min at ambient temperature. The absorbance was read at 595 nm using a DR6000 UV–Vis spectrophotometer (HACH, Düsseldorf, Germany). The protein content was calculated based on a standard curve made with bovine serum albumin with concentrations ranging between 0.1 and 1.4 mg/mL.

The protein content of the FPIs was extracted following Mæhre et al. [57]. About 1 g sample was homogenized with 60 mL of 0.1 N NaOH in 3.5% NaCl solution. The homogenates were then incubated in a water bath at 60 °C for 90 min, which was followed by centrifugation at 4 °C for 30 min at 5000 rpm (TJ-25 Centrifuge, Beckman Coulter, Brea, CA, USA). The supernatants were measured for protein content using the Bradford method as described above.

#### 2.2.5. Antioxidant Activities

Antioxidant activities of the FPHs were evaluated through 2,2-diphenyl-1-picrylhydrazyl radical scavenging activity (DPPH-RSA) based on the method described by Fu et al. [58]. The total reducing power capacity (TRPC) of the FPHs was measured using the method described by Oyaizu [59].

#### Determination of DPPH-RSA

Approximately 2.0 mL of FPH sample was mixed with 1 mL of 96% ethanol and 1 mL of DPPH solution (0.1 mM in 96% ethanol). The reaction tubes, in triplicate, were wrapped in aluminium foil, shaken well, and then left to incubate for 30 min at room temperature. The absorbance of the solutions was read at 517 nm (DR6000 UV–Vis, Hach, Düsseldorf, Germany). For the control sample, 2 mL of 96% ethanol was used instead of the FPH sample. The DPPH-RSA was calculated using Equation (4):
(4)$$\mathrm{DPPH}-\mathrm{RSA}\;(\%)=\frac{A_{\mathrm{control}}-{\;A}_{\mathrm{sample}}}{A_{\mathrm{control}}}\times 100$$
where A_control_ and A_sample_ are the absorbances of the control and sample solution read at 517 nm, respectively.

#### Determination of TRPC

Exactly 2 mL of FPH sample was mixed with 2 mL of 0.2 M phosphate buffer (pH 6.6) and 2 mL of 1% potassium ferricyanide. The mixture was then incubated for 20 min at 50 °C. After that, 2 mL of 10% TCA was added to the mixture. Approximately 2 mL of the incubated mixture was mixed with 0.4 mL of 1% ferric chloride and 2 mL of distilled water in a test tube. The absorbance of the solution was read at 700 nm after a 10 min reaction (DR6000 UV-VIS, Hach, Düsseldorf, Germany). Each sample was measured in triplicate. A standard curve was made using a standard vitamin C solution spanning a concentration range of 0 to 20 μg/mL, and the TRPC was expressed as equivalent (equiv.) mg vitamin C/g FPH protein.

High DPPH-RSA and TRPC values indicated that the FPHs had high antioxidant activities.

#### 2.2.6. Statistical Analysis

All data summaries and statistical analyses were carried out in Microsoft Excel 365 (Microsoft Inc., Redmond, WA, USA) and IBM SPSS Statistics software (Version 22, IBM, 1 New Orchard Road, Armonk, New York, NC 10504-1722, United States). One-way analysis of variance (ANOVA) and Tukey’s HSD tests were performed on the means of each variable. Significant levels were defined as *p* < 0.05 for all statistical analyses.

## 3. Results and Discussion

### 3.1. Proximate Composition of the Raw Materials and FPIs

The crude protein content was significantly higher, and the lipid and ash contents significantly lower, in the FPIs than in the corresponding raw materials (Table 1). Thus, the alkaline pH-shift process effectively removed insoluble impurities (bone, skin, and connective tissues) and lipids from the FPIs. Nguyen et al. [12] found that over 90% of the total lipids and 85% of the total ash in the raw materials were removed during FPI processing. Lipid removal during FPI production is advantageous for further processes, since muscle lipids are highly susceptible to oxidation, increasing the risk of the formation of rancidity [42,60,61]. Other common pro-oxidants in the side streams, such as myoglobin (Mb), haemoglobin (Hb), and iron, can also be removed successfully during FPI processing [12,42], thus reducing the risk of oxidation even further. However, the pro-oxidants present in the substrate may react with antioxidative peptides generated during and after hydrolysis. Lowering these pro-oxidant compounds may thus preserve the antioxidant properties of the FPHs obtained [62,63].

Lowering the lipid content from the lipid-rich side-streams through FPI processing is beneficial, and it is in agreement with Kristinsson and Rasco [18], who suggested that excess lipids must be removed from lipid-rich raw materials before they are used for FPH production. Halldórsdóttir et al. [64] also reported applying the pH-shift method to recover proteins and remove undesirable components from saithe (*Pollachius virens*) mince. The hydrolysis process in the Halldórsdóttir et al. [64] study resulted in higher quality FPHs than when processed the traditional way without dewatering. In addition, Khantaphant, et al. [63] observed that lowering the lipid content and impurities by applying membrane separation followed by washing to brownstripe red snapper (*Lutjanus vitta*) mince before hydrolysis can form FPHs with higher antioxidative activities than FPHs prepared directly from the original mince.

### 3.2. Optimization of FPH Processing from DM-FPIs

#### 3.2.1. Effects of Enzyme Ratios (Trial 1)

The degree of hydrolysis (DH) is used as an indicator for peptide bond cleavage, while protein recovery (PR) indicates the yield obtained from the hydrolysis process. The DH can affect the PR and other functional properties, such as antioxidative activities [65,66]. Overall, the DH and PR increased when the enzyme–protein substrate ratio increased from 0 to 3.0% (Figure 2A,B). During the hydrolysis process (i.e., with Alcalase), the enzyme divides the peptide bonds in the initial proteins into smaller protein molecules and peptides with higher solubility [67]. The hydrolytic enzyme breaks down the peptide bonds of the protein substrates through their active sites, which may initiate catalysis through covalent interaction with the protein substrates [68]. Therefore, increasing the enzyme concentration speeds up the reaction, as long as there is enough substrate available to bind to, resulting in an increase in both the DH and PR. The PR and antioxidant activities significantly increased when the 0.5% enzyme was added compared to the autolysis of the substrate (no enzyme added). However, the DH was not significantly different with these two enzyme ratios (*p* > 0.05). When no exogenous enzyme was added, the DH was 14.1%. This value was higher than the DH from the autolysis of Tra catfish side-streams under the same hydrolysis conditions (i.e., temperature of 50 °C for 3 h) studied by Nam et al. [14], who obtained a value of 6.5%. In the autolysis of yellowfin tuna (*Thunnus albacares*) performed for 3 h at 45 °C, the resulting DH was about 5% [69]. These results may reflect the fact that the hydrolysis had already occurred in the substrate (i.e., DM-FPI) in this study, as discussed by Nguyen et al. [12]. There was no significant difference in the DH when the enzyme ratios ranged from 1.0% to 2.5%. However, the DH significantly increased when the enzyme ratio exceeded 2.5%, and the highest DH value of 33.3 ± 4.0% was obtained at an enzyme ratio of 3.5%. This result was in agreement with the production of FPHs from Tra catfish by-products using Alcalase studied by Nam et al. [14], where the highest DH (30.7%) was obtained at the enzyme–protein substrate ratio of 3.4%. The increase in the DH with higher enzyme ratios has been shown in early studies [14,43,44,69]. However, in the current study, the DH only decreased slightly when the enzyme concentration exceeded 3.5%. Moreover, some of the peptides generated in the FPH may be further hydrolysed, forming free amino acids and smaller peptides when the enzyme ratio increases, leading to a decline in the DH [70].

The PR increased significantly, from 8.0% to 58.1%, when the enzyme ratio increased from 0% to 3.0%. Afterwards, the PR remained stable when the enzyme ratio was increased to 5% (*p* > 0.05). Nam et al. [14], who prepared FPHs from a mixture of Tra catfish side-streams using the enzyme Alcalase, also found that the highest nitrogen recovery (82.2%) was obtained at an enzyme–protein substrate ratio of 3.4%. This may have been due to saturation of the existing peptide bonds, especially soluble peptides in the hydrolysis solution, and thermal denaturation of the enzyme [14,65].

DPPH is a comparatively stable radical used as a substrate to determine antioxidant efficacy [71]. The DPPH-RSA increased slightly, from 22.0% to 38.5%, when the enzyme ratio increased from 0.5% to 2.5% (Figure 2C). Furthermore, there was a significant increase in DPPH-RSA when the enzyme ratio exceeded 2.5%, and the DPPH-RSA reached the highest value (79.8 ± 6.9%) when the enzyme was used at a concentration of 3.5%. These results reveal that the FPHs studied possibly contained amino acids or peptides, which can act as electron donors, reacting with free radicals to improve the stability of products and terminate radical chain reactions. The increase in DPPH-RSA when the enzyme ratio increased from 0.5% to 3.5% may have been due to the increase in the extent of hydrolysis at higher enzyme concentrations, releasing more antioxidative peptides, which are normally inactive within the sequence of the precursor protein molecules [72]. However, the DPPH-RSA significantly decreased when the enzyme ratio exceeded 3.5%. This may have been due to the breakdown of the already formed antioxidative peptides during the early stages of the hydrolysis process. Similar results were observed by Tanuja et al. [73], who showed a reduction in the DPPH-RSA of the FPH produced from Tra catfish frame meat when the Alcalase concentration (% *v*/*w* of substrate protein) was increased from 0.5% to 2.5%.

The reducing capacity of a compound may serve as an indicator of its antioxidant activity. The presence of reducers (i.e., antioxidants) leads to a reduction of the Fe^3+^/ferricyanide complex to the ferrous form (Fe^2+^). In this assay, the yellow colour of the test mixture changed to various shades of blue and green, depending on the reducing power of each sample. Therefore, measurement of the formation of Perl’s Prussian blue at 700 nm can be used to monitor the Fe^2+^ concentration. In this study, the TRPC had a similar change trend as the DPPH-RSA, with increases ranging from 0.74 equiv. mg vitamin C/g FPH protein to 2.0 equiv. mg vitamin C/g FPH protein when the enzyme ratio increased from 0 to 2.5% (Figure 2D). The hydrolysis of DM-FPI (as expressed by the increased DH) into peptides may give rise to both the release of sequences with antioxidant properties and the exposure of previously hidden amino acid residues and side chains with antioxidant activity, as discussed above, resulting in an increase in TRPC. No significant change was observed in the TRPC as the enzyme concentration was increased from 3.0−5.0%. In this trial, the changes in the antioxidant activities and the DH showed similar trends. This result was consistent with the study by Klompong et al. (2007), who showed that the TRPC increased with an increased DH.

Based on the results of trial 1, the enzyme–substrate protein ratio of 3.0 was chosen as the optimum enzyme ratio for the FPHs produced from the DM-FPI.

#### 3.2.2. Effects of Hydrolysis Temperature (Trial 2)

The hydrolysis temperature had significant effects on the DH (Figure 3A). The DH slightly increased when the temperature increased from 45 °C to 50 °C. The DH then increased sharply, from 17.0% to 43.2%, when the hydrolytic temperature increased from 50 °C to 65 °C. However, the DH significantly dropped to 20.1% when the temperature reached 70 °C. According to Eisenthal et al. [74], temperature influences the stability and activity of enzymes. As mentioned above, the hydrolysis reaction is based on the covalent interaction between specific enzyme groups (active groups) and the protein substrate. Hence, this reaction rate depends on the enzyme’s specific three-dimensional structure [68]. In addition, the increased hydrolysis temperature may increase the internal energy of the enzyme [75]. Enzyme activity increases with temperature as long as the enzyme is stable, which resulted in increases in the DH, especially when the temperature increased from 50 °C to 65 °C. However, the enzyme is a protein and may be denatured and inactivated at high temperature [76] (in this study, when the temperature was above 65 °C), resulting in a decreased DH. These results were in agreement with an observation by Amiza et al. [77], who found that the effect of temperature on the DH of Tra catfish frame hydrolysis showed a bell-sharped pattern. Below the optimal temperature, the DH increased. However, above the optimal temperature value, the DH was reduced due to enzyme denaturation and inactivation. In addition, the enzyme catalyses rapidly with the insoluble protein molecules and then polypeptide chains that are poorly bonded to the surface are hydrolysed. The proteins that are cleaved the slowest are the more compacted core proteins [44]. At the temperature of 70 °C, the substrate proteins may be partly aggregated and less susceptible to enzymatic hydrolysis, leading to a lower DH than at lower temperatures.

Several authors have noted a positive correlation between the DH and the PR during FPH production. However, in this study, although the DH strongly increased when the reaction temperature was increased from 45 °C to 65 °C, the increase in the PR was not statistically significant, with values ranging from 54.1% and 61.4% (Figure 2B). This may have been due to the initial substrate being partly hydrolysed before entering the FPH production process, as discussed above. The initial substrate used in this study was obtained with the pH-shift method and mainly composed of sarcoplasmic and myofibrillar proteins, some of which were even partly hydrolysed, resulting in high solubility [12]. Therefore, the initial substrate had high solubility that could be extracted to the FPHs even at a low DH. The PR only significantly decreased when the temperature reached 70 °C (41.3%), which was consistent with the DH reduction at this temperature. This may have been because the enzyme was thermally denatured, leading to lower activity [65].

The DPPH-RSA significantly increased, from 57.0% to 75.1%, when the hydrolysis temperature increased from 45 °C to 50 °C (*p* < 0.05) (Figure 3C). The enzymatic hydrolysis led to the release of antioxidative peptides, which function as free radical scavengers. This may have been due to the higher enzyme activity obtained at the higher temperatures, resulting in more hydrolysis. The increase in hydrolysis may have resulted in a higher content of low-weight molecular peptides, which are the main components related to radical scavenging activity [78,79]. However, there was a significant reduction in the DPPH-RSA when the temperature continued to increase to 65 °C. These findings are comparable with those from a study on mackerel (*Pneumatophorus japonicus*) FPH production with the use of Alcalase that showed that the highest DPPH-RSA was obtained when the hydrolysis was performed at a temperature of 46 °C, which was followed by a significant reduction when the temperature increased to 60 °C [80]. The DPPH-RSA changes were the opposite of the DH changes and this relationship differed from the correlation between these two parameters in trial 1. This reflects the fact that, after obtaining the highest DPPH-RSA value, the excess hydrolysis (expressed by a higher DH) could break the released antioxidative compounds in the FPH, resulting in a decrease in the DPPH-RSA. This finding is in agreement with those of Klompong et al. [44], who showed that, as the DH increased, from 5% to 25%, the antioxidant activities of the FPH produced from yellow stripe trevally (*Selaroides leptolepis*) with Alcalase decreased. Furthermore, these antioxidative compounds may be sensitive to temperature, since oxidation occurs more rapidly at higher temperatures.

The TRPC changes were not completely consistent with the DPPH-RSA changes in this trial (Figure 3D). This may have been because the peptide contributors for these two antioxidant activities differed. The TRPC was not significantly different in the hydrolysis temperatures ranging between 45 °C and 55 °C (*p* > 0.05). There was a slight decrease when the hydrolysis temperature was increased to the temperature range of 60 °C to 65 °C. This reduction in the TRPC when the DH obtained the maximum value may have been due to the breakdown of ferric reducing agents at the stage of excessive hydrolysis, similarly to the DPPH, as discussed above. Therefore, the highest TRPC was obtained (2.6 ± 0.3 equiv. mg VTM C/g FPH protein) when the degree of hydrolysis was limited at the temperature of 70 °C.

The optimum temperature for the hydrolysis of DM-FPI with Alcalase in this study was 50 °C. This is consistent with the FPH production from Tra catfish frames with Alcalase studied by Amiza et al. [81]. Wasswa et al. [82] also reported an optimum temperature of 50 °C for grass carp skin (*Ctenopharyngodon Idella).* However, other studies show different optimal temperatures for Alcalase activity in FPH production, indicating that the optimal temperature is species-dependent, as well as being dependent on different substrate and reaction conditions [65,83,84].

#### 3.2.3. Effects of Hydrolysis Time (Trial 3)

DH increased, from 18.9% to 31.5%, as the hydrolysis time was increased from 1.5 to 9.0 h (*p* < 0.05) (Figure 4A). This was consistent with the findings from a study on FPH production from Tra catfish by-products [14] that showed an increase in the DH when the time was increased up to 15 h. Other studies on other fish species have also shown the same positive relationship between hydrolysis time and the DH, including species such as Pacific whiting (*Merluccius productus*) [65], silver carp (*Hypophthalmichthys molitrix*) [39], and skipjack tuna (*Katsuwonus pelamis*) [85].

PR did not change much, although the DH changed significantly with time (Figure 4B). This may have been due to the fact that the FPHs were produced from FPI and any impurities containing insoluble proteins had already been removed, leading to higher concentrations of extractable proteins in the initial substrate. The PR remained constant during the first 4.5 h, reaching a maximum value at a hydrolysis time of 6 h (67.5 ± 2.0%), and decreased when the hydrolysis was extended to 9 h. This decrease may have been due to the further degradation of peptides to free amino acids and/or other volatile compounds [79]. The highest PR in this study was lower than that obtained by Nam et al. [14] (81.9%) for FPHs produced from Tra catfish by-products using Alcalase. This difference is likely due to the differences in evaluation methods, raw materials, and hydrolysis conditions.

DPPH-RSA significantly increased, from 57.0% to a maximum value of 75.1%, when the hydrolysis was prolonged from 1.5 to 3.0 h (Figure 4C). This may have been due to the increase in hydrolysis over this period, resulting in greater amounts of compounds with DPPH radical scavenging potential. Oxidative processes may occur during hydrolysis [62]. Therefore, extending the hydrolysis beyond 3.0 h decreased the DPPH to 48.9% for 6 h hydrolysis. However, the DPPH-RSA increased when the hydrolysis was further extended to 7.5 h, followed by a reduction after that. These fluctuations reflected a possibility that two concurrent processes may have affected the DPPH-RSA during enzymatic hydrolysis. The first process was the release of the antioxidative peptides and the second was the breakdown of the generated antioxidative peptides. Dong et al. [39] found that the DPPH-RSA of FPHs was affected by the hydrolysis time, with the highest value obtained after 2 h of hydrolysis and a slight reduction after 6 h of hydrolysis. Wang et al. [80] studied FPH production from mackerel (*Pneumatophorus japonicus*) and showed that the DPPH-RSA increased with time, obtaining a maximum value after 5 h followed by a decline when the hydrolysis was prolonged to 8 h.

Similarly to the DPPH-RSA, the TRPC significantly increased, from 1.5 equiv. mg vitamin C/g FPH protein to 2.3 equiv. mg vitamin C/g FPH protein, when the hydrolysis was prolonged from 1.5 to 3 h (Figure 4D). However, there was a significant decline in the TRPC when the hydrolysis was extended to 9 h down to 1.2 equiv. mg vitamin C/g. This may have been due to further oxidative processes related to the antioxidant peptides released in the FPHs during hydrolysis, as observed by Halldorsdottir, et al. [62]. Furthermore, the excessive hydrolysis (with an increased DH) after 3 h may have degraded the obtained antioxidative peptides and led to a reduction in the TRPC [70].

A hydrolysis time of 3 h was thus regarded as optimal for the production of FPHs with high PR and antioxidant activities. This was similar to the value obtained by Amiza et al. [81], who indicated that 163 min was the optimal reaction time for the hydrolysis of Tra catfish frames with Alcalase. The optimal hydrolysis time for the preparation of FPHs from visceral waste proteins of Catla (*Catla catla*) with Alcalase was 135 min, as indicated by Bhaskar et al. [86].

### 3.3. Comparison of the FPHs Prepared from Different FPIs

#### 3.3.1. Amino Acid Composition

The amino acid compositions of food proteins play important roles in various physiological activities of the human body. They also relate to foods’ functional roles and potential within the food industry [72]. All the FPHs studied had similar amino acid compositions (Table 2). The amino acid compositions of the FPHs were similar to those in the corresponding initial substrates (FPIs), which were reported by Nguyen et al. [12]. However, the hydrophobic proteins were lower in the FPHs compared to the corresponding FPIs. This may have been because the hydrophobic proteins remained in the sediment after the production of the FPHs.

Furthermore, gamma-aminobutyric acid and cysteine were observed in all FPHs, although these amino acids were not detected in the corresponding substrates [12]. The presence of the cysteine in the FPHs may have been because cystine was reduced to form cysteine during hydrolysis, as indicated by the lower cystine content in the FPHs compared to the FPIs. Glutamic acid, aspartic acid, lysine, and leucine were the main components of all FPHs, similarly to FPHs produced from capelin (*Mallotus villosus*), Pacific whiting (*Merluccius productus*), and herring (*Clupea harengus*) [72]. The amino acid compositions of all the FPHs met the human amino acid requirements for adults recommended by the FAO/WHO/UNU [87], indicating that these FPHs may be useful as additives or ingredients when developing products for adults, providing valuable input to a balanced protein diet [72].

#### 3.3.2. DH, PR, and Antioxidant Activities

The DH was not significantly different between the three FPHs produced from different side streams (*p* > 0.05). However, the PR was highest in the ACO-FPH (80 ± 6.3%) compared to 58.4 ± 2.0% in the DM-FPI and 60.4 ± 5.7% in the HBB-FPH (Table 3). This may have been due to the ACO-FPH’s initial substrate (ACO-FPI) having higher soluble protein content than the DM-FPI and HBB-FPI [12].

The antioxidant activity of peptides is closely related to their amino acid constituents and their sequences [78]. In this study, all the FPHs had similar amino acid profiles (Table 2). However, their antioxidant activities were significantly different. Therefore, the peptide sequences of the FPHs may play a primary role in the antioxidant activities. Furthermore, the pro-oxidant content in the initial substrates (myoglobin, heme, iron, etc.) may have also affected the antioxidant activities of the obtained FPHs. The HBB-FPI may have contained higher amounts of pro-oxidants, such as myoglobin and iron [12], which could have reacted with the antioxidant peptides during hydrolysis, resulting in the lowest bioactivity values, as reflected by the DPPH (57.9 ± 3.6%) and TRPC (2.0 ± 0.2 equiv. vitamin C/g FPH protein), respectively in the HBB-FPH. In contrast, the ACO-FPI contained lower amounts of pro-oxidants and, correspondingly, resulted in the ACO-FPH having the highest DPPH (86.1 ± 1.6%) and TRPC (6.4 ± 0.4 equiv. mg vitamin C/g FPH protein). The radical scavenging activity was found to correlate with the Met, Arg, Val, His, Pro, and Asp content in the peptide sequences of the FPHs [71,88]; when combined, these amino acids reached a significant amount (about 25 g/100 g protein) in all FPHs (Table 2). Overall, the results suggest that the FPHs from Tra catfish side-streams have good potential as natural antioxidant ingredient in foods.

## 4. Conclusions

The present study clearly indicated that the conditions of hydrolysis with Alcalase, including the enzyme concentration, temperature, and hydrolysis time, had significant effects on the properties of the FPHs produced from the Tra catfish dark muscle FPIs using the pH-shift method. An enzyme ratio of 3.0%, hydrolysis temperature of 50 °C, and hydrolysis time of 3 h were established as the optimal hydrolysis conditions, as the highest antioxidant activities (DPPH-RSA and TRPC) and relatively high protein recovery were obtained at these conditions.

The FPHs produced from DM-FPI, HBB-FPI, and ACO-FPI had different chemical properties. These materials should thus be processed into FPHs separately, adjusting each FPI towards the production of a specific value-added food product. However, all FPHs showed high antioxidant activities, especially the ACO-FPH and DM-FPH, which had high DPPH-RSA and TRPC, indicating the great potential of these FPHs as food antioxidants. Furthermore, the obtained FPHs showed high nutritional value with significant essential amino acid contents, indicating that the FPHs could also be used as food supplements.

However, other physicochemical properties, such as solubility, foaming ability, water- and oil-holding capacity, emulsifying properties, and lipid stability, should be studied further to shed more light on the potential application range of FPHs. In addition, FPHs may contain various types of peptides, which have specific bioactive properties and different bioavailability. Therefore, peptide groups from these FPHs should be purified further, and their properties should be identified to obtain more information about the bioactive and bioavailability characteristics of each peptide.

## Figures and Tables

**Figure 1 foods-11-04102-f001:**
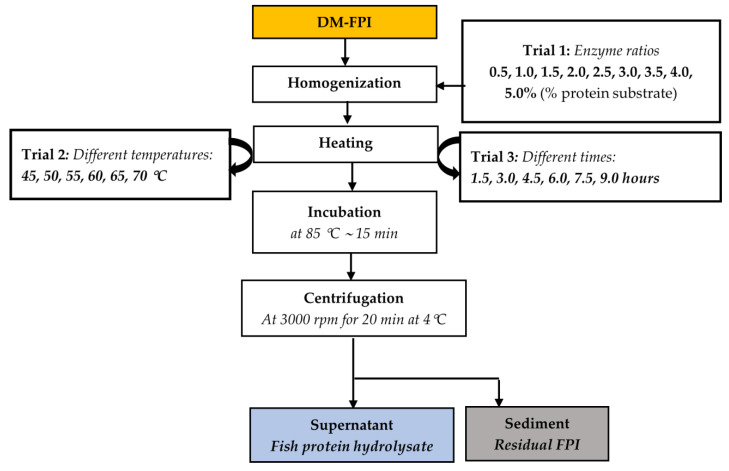
Experimental design for the optimization of FPH production from Tra catfish DM-FPI (dark muscle fish protein isolate) performed with three trials. Orange-filled box: protein, water-lipid, and ash content measured; blue-filled box: weight of the protein solution recorded and its protein content determined; grey-filled box: weight recorded and water content determined.

**Figure 2 foods-11-04102-f002:**
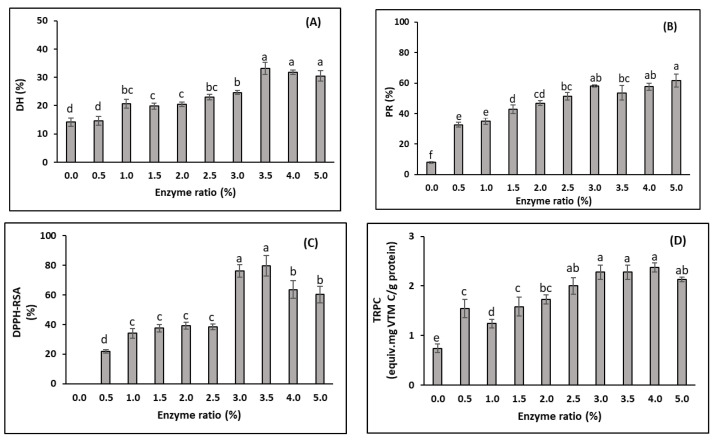
Effects of the enzyme/substrate ratios on properties of FPHs obtained from DM-FPI (trial 1), including: (**A**) degree of hydrolysis (DH, %); (**B**) protein recovery (PR, %); (**C**) DPPH radical scavenging activity (DPPH-RSA, %); and (**D**) total reducing power capacity (TRPC, equiv. mg vitamin C/g FPH protein). The hydrolysis reaction was carried out at 50 °C for 3 h. Different lowercase letters show significant differences at a significance level of *p* < 0.05.

**Figure 3 foods-11-04102-f003:**
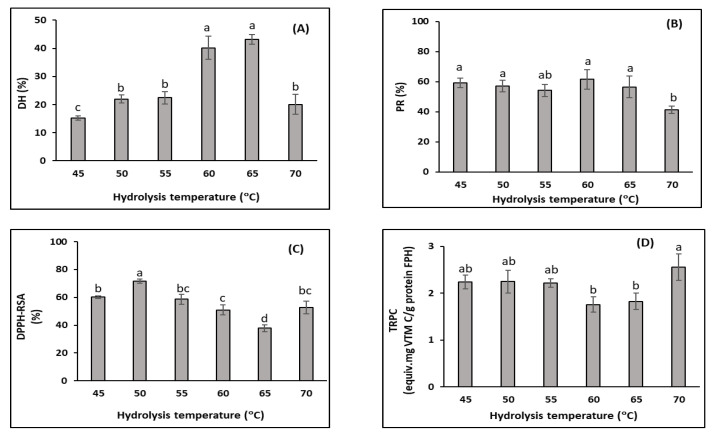
Effect of hydrolysis temperature (°C) on FPH properties from the DM-FPI (trial 2), including: (**A**) degree of hydrolysis (DH, %); (**B**) protein recovery (PR, %); (**C**) DPPH radical scavenging activity (DPPH-RSC, %); and (**D**) total reducing power capacity (TRPC, equiv. mg vitamin C/g FPH protein). The hydrolysis reaction was carried out for 3 h with an enzyme ratio of 3%. Different lowercase letters show significant differences at a significance level of *p* < 0.05.

**Figure 4 foods-11-04102-f004:**
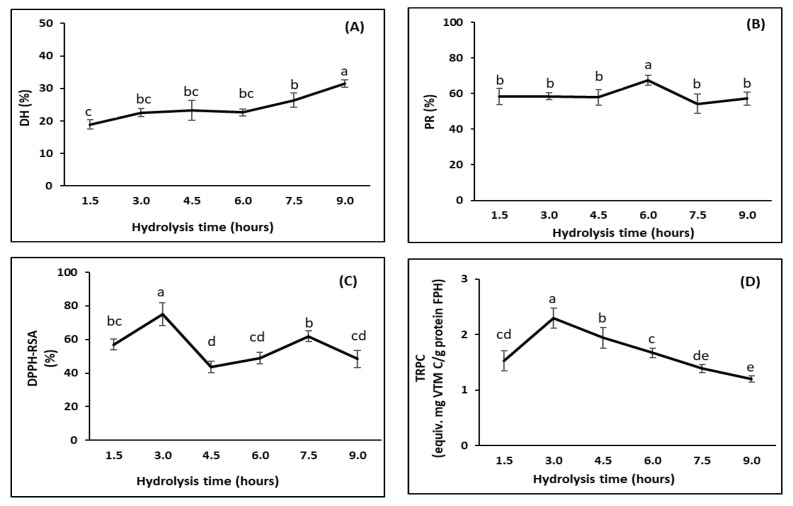
Effect of hydrolysis time (hours) on characteristics of FPHs obtained from the DM-FPI (trial 3), including: (**A**) degree of hydrolysis (DH, %); (**B**) protein recovery (PR, %); (**C**) DPPH radical scavenging activity (DPPH-RSC, %); and (**D**) total reducing power capacity (TRPC, equiv.mg vitamin C/g FPH protein). The hydrolysis reaction was carried out at 50 °C with an enzyme ratio of 3%. Different lowercase letters show significant differences at a significance level of *p* < 0.05.

**Table 1 foods-11-04102-t001:** Proximate composition (%) of the fish protein isolates (FPIs) produced from Tra catfish dark muscle, head and backbone blend (HBB), and abdominal cut-offs (ACOs) *****. Results are expressed as means ± SD from triplicate measurements (*n* = 3) **.

		Water Content	Crude Protein	Lipid Content	Ash Content
Raw material	Dark muscle	66.5 ± 1.0 ^A^	14.7 ± 0.2 ^B^	17.6 ± 1.5 ^B^	0.8 ± 0.1 ^B^
	HBB	54.8 ± 1.0 ^C^	15.2 ± 0.1 ^A^	21.9 ± 0.6 ^A^	7.8 ± 0.4 ^A^
	ACO	60.8 ± 2.0 ^B^	15.3 ± 0.1 ^A^	23.5 ± 1.1 ^A^	1.0 ± 0.1 ^B^
FPIs	DM-FPI	73.9 ± 0.7 ^ab^	23.5 ± 0.9 ^a^	3.2 ± 0.0 ^a^	0.1 ± 0.0 ^a^
	HBB-FPI	77.3 ± 0.8 ^a^	20.4 ± 0.5 ^b^	2.8 ± 0.4 ^a^	0.1 ± 0.0 ^a^
	ACO-FPI	73.1 ± 2.1 ^b^	24.4 ± 1.4 ^a^	3.1 ± 0.1 ^a^	0.0 ± 0.0 ^a^

* The data were adapted from Nguyen et al. [12]. ** Different uppercase letters indicate significant differences within the column for the raw material (A, B, C); different lowercase letters show significant differences within the column for the FPI products (a, b) at a significance level of *p <* 0.05.

**Table 2 foods-11-04102-t002:** Amino acid compositions (g/100 g protein) of the fish protein hydrolysates produced from fish protein isolates prepared from dark muscle (DM-FPH), head and backbone blend (HBB-FPH), and abdominal cut-offs (ACO-FPH).

Amino Acids	DM-FPI **	HBB-FPI **	ACO-FPI **	DM-FPH	HBB-FPH	ACO-FPH	FAO/WHO/UNU *
4-Hydroxyproline	ND	ND	0.6	ND	ND	ND	
Alanine ^b^	6.3	6.5	6.4	6.6	6.5	6.6	
Arginie ^a^	6.6	6.9	6.3	6.4	5.9	6.6	
Aspartic acid	11.2	12.1	10.5	10.7	10.7	10.9	
Cystein	ND	ND	ND	0.6	0.7	0.6	
Cystine	1.1	1.1	1.0	0.7	0.6	0.7	
Gamma-aminobutyric acid	ND	ND	ND	0.5	0.6	0.6	
Glutamic acid	18.4	19.4	17.2	20.4	19.2	20.4	
Glycine ^b^	3.7	3.9	5.1	3.3	3.4	3.4	
Histidine ^a^	2.6	2.6	2.8	2.4	2.5	2.2	1.5
Isoleucine ^ab^	4.7	5.0	4.6	4.0	4.2	3.9	3.0
Leucine ^ab^	8.8	9.1	7.9	8.5	8.8	8.5	5.9
Lysine ^a^	9.3	10.1	8.8	10.2	9.6	10.4	4.5
Methionine ^ab^	3.3	3.5	2.9	2.8	2.8	3.2	2.2
Phenylalanine ^ab^	3.8	4.2	3.8	3.1	3.1	2.9	3.8
Proline ^b^	4.6	4.3	5.0	3.6	4.0	3.4	
Serine	4.2	4.4	4.1	4.5	4.8	4.6	
Threonine ^a^	4.6	4.9	4.2	4.5	4.5	4.4	2.3
Tyrosine	2.2	3.5	3.1	2.6	2.8	2.7	
Valine ^ab^	5.0	5.3	4.8	4.3	4.8	4.4	3.9
Total amino acids	100.5	106.8	99.1	99.6	99.5	100.2	
Total essential amino acids	48.8	51.7	46.2	46.2	46.3	46.4	
Total hydrophobic amino acid	40.3	41.8	40.6	36.3	37.6	36.2	

^a^ Essential amino acid for infants. ^b^ Hydrophobic amino acids. * FAO/WHO/UNU recommendations for adults [87]. ** The data were adapted from Nguyen et al. [12].

**Table 3 foods-11-04102-t003:** Efficiency of the fish protein hydrolysate (FPH) production from fish protein isolates obtained from Tra catfish dark muscle (DM-FPH), head and backbone blend (HBB-FPH), and abdominal cut-offs (ACO-FPH). Results are expressed as means ± SD from triplicate measurements (*n* = 3) *.

Production	DH (%)	PR (%)	DPPH (%)	TRPC (Equiv. mg Vitamin C/g FPH Protein)
DM-FPH	22.5 ± 1.3 ^a^	58.4 ± 2.0 ^b^	75.1 ± 6.8 ^a^	2.3 ± 0.2 ^b^
HBB-FPH	22.9 ± 1.6 ^a^	60.4 ± 5.7 ^b^	57.9 ± 3.6 ^b^	2.0 ± 0.2 ^b^
ACO-FPH	24.0 ± 3.5 ^a^	81.5 ± 4.3 ^a^	86.1 ± 1.6 ^a^	6.4 ± 0.4 ^a^

* Different superscript letters indicate significant differences within each column at *p* < 0.05.

## Data Availability

Data are contained within the article.

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
