# Peer review of "Protein Characteristics and Bioactivity of Fish Protein Hydrolysates from Tra Catfish (Pangasius hypophthalmus) Side Stream Isolates"

_foods, 2022, doi:10.3390/foods11244102_

Round 1

Reviewer 1 Report

This manuscript investigated the feasibility of obtaining functional fish protein hydrolysates (FPHs) by enzymatically hydrolysing FPIs. Suggestions for modification are as follows:

Line 53, What’s fish side streams?

Line 94, What’s pH- shift method?

Line 119, Why the pH is 5.5?

Line 121-122, “obtained proteins were washed with distilled deionized water to a neutral pH”. Can the protein be washed to neutral pH with water?

Line 123, How to remove the NaCl?

Line 124, How to get 50g sample?

Line 276-278, “such as myoglobin (Mb), haemoglobin (Hb), and iron, may also be removed successfully during FPI processing”. The author did not do these tests, why put this reference in it?

Line 303-304, Could the author explain why the DH and PR increased when the enzyme-protein substrate ratio increased?

Line 325-326, Could the author explain why the PR remained stable when the enzyme ratio was increased to 5%?

Line 353-367, Could the author explain why the TRPC increased with increased DH?

Line 371-375, Could the author explain why the DH slightly increased when the temperature increased from 45℃ to 50℃. The DH then increased sharplywhen the hydrolytic temperature increased from 50℃ to 65℃ and the DH significantly reduced when the temperature reached 70℃?

Line 441-442, Could the author explain why DH increased, from 18.9 ± 1.4 % to 31.5 ± 1.1%, as the hydrolysis time was increased from 1.5 to 9.0 hours?

Line 532-533, Could the author explain why all the FPHs had a similar amino acid profile ,However, their antioxidant activities were significantly different?

Reviewer 2 Report

In the Abstract, the aims of this study are missing

Authors should add brief methodology in abstract

Please add quantitative results in the abstract section.

Add proper conclusive line in the end of abstract that should summarize the whole study.

Line 31. Remove ‘catfish’ and “fish protein hydrolysates” fish protein hydrolysates from keywords as it is already used in title

Line 44 Please mention how fish can be helpful in elimination of food security.

Add rationale of the study in the end of introduction that should highlight the importance and reasoning of the study.

In section 2.1.1. Please use full form of FPI it may cause confusion to the reader.

In line 173 and some other methods can you please provide the method of ISO or please add manual of these methods in supplementary file.

In Line 199 please provide the sample preparation method of LC-MS.

Mention how bioavailability was measured?

Statistical analysis are not clear please elaborate comprehensively for each parameter.

In proximate results table why small and capital lettering has been used if you want to use it please denote them in the end of the table.

In line 293 please correct the citation and convert it in numbering.

Discussion is not clear please give proper comparison along with justification and previous references.

In title there is bioavailability, but there is nothing about it in the whole article, please mention about it.

Line 530. Describe lettering in table below the table

Please revise your conclusion there is no sequence. Also summarize it and be concise.

Please correct the reference no 1 that is not according to the format.

Please go through the whole article and improve its grammar.

Please recheck your all references according to the journal format.   

Reviewer 3 Report

Hi dear

This article "Protein characteristics and bioactivity of fish protein hydrolysates from Tra catfish (Pangasius hypophthalmus) side stream isolates” was revised and has a novelty and I recommend it for publication after consideration of the following major comments.

·       Please include background of Tra catfish in terms of composition or meat chemical properties.

·       Line 20: pH-shift method (FPIs)? Abbreviation is corrected.

·       Line 22: “different substrates” is complexity for reader in the first time of reading. What is the optimal hydrolysis degree for your target treatment?

·       Line27: “6.4 equiv. mg vitamin C/g protein”. Please correct it as scientific expression.

·       The type of statistical design used in this research should be mentioned.

·       Please state the results in an orderly manner and in full detail in the abstract along with the comparative significance level error.

·       Why alkalase was used in your research, state the reason in the abstract and of course in the introduction of the article.

·       Line 72-82: you can refer to the new following article: (DOI: 10.1111/jfpp.15456)

·       Line 102-106: Please include the study treatments (DM-FPI, HBB-FPI and ACO-FPI), statistical design, and the name of the responses assessed.

·       In your research, why didn't you investigate the toxicity and pathogenicity of the final product? Because these two attributes can be important in marine products.

·       Line 126: “Preparation of FPH from the DM-FPI” How about the HBB-FPI and ACO-FPI? Please point it to them.

·       Fig 1: What is the basis for choosing the best conditions for trials 1, 2 and 3? What were the best results for you for each trial and why?

·       Table 1: for water content in “DM-FPI” namely 73.9 ± 0.7bc, I think statistical symbol is not corrected (maybe ab instead of bc etc.).

·       Fig 2: All Tables and Figures: The alphabetical statistical letters for the means should all be modified such that the greatest number has the letter a and as the numbers go lower, letters b, c etc.

·       Line 372-373 etc: The standard error or deviation should be deleted.

·       Explain why you did not study the combined effects of enzyme concentration, temperature and enzyme action time. Do you think your method is scientific? And you were stating an optimal treatment. Definitely, these three investigated factors together have interfering and controlling effects.

·       Table 2: It is better to make a statistical comparison for each amino acid in a row

·       Discussion text must grammar improve and in some cases it is very weak and maybe there is no discussion at all.

·       Conclusion is very general, try to make it more scientific, comprehensive and concise in detail, especially.

Round 2

Reviewer 2 Report

Authors have summarized well, it can be accepted now.

Reviewer 3 Report

Hello dear authors

It seems that the quality of the article has become much better with the proper implementation and correction of the suggested comments and it has the ability to be published in the journal.